# How do befriending interventions alleviate loneliness and social isolation among older people? A realist evaluation study

**Olujoke A. Fakoya** [1,2]*, **Noleen K. McCorry**[1,2], **Michael Donnelly**[1,2]

**1** Centre for Public Health, Queen's University Belfast, Belfast, Northern Ireland, **2** Centre of Excellence for Public Health, CoE, Queen's University Belfast, Belfast, Northern Ireland

* ofakoya01@qub.ac.uk

## Abstract

### Background

Befriending is a popular way in which to intervene to combat loneliness and social isolation among older people. However, there is a need to improve our understanding about how these interventions work, for whom and in which contexts, to make the best use of the increasing investment in the provision and delivery of befriending services.

### Methods

A realist evaluation was undertaken as it focuses on uncovering causal processes and interactions between mechanisms and contextual characteristics. Five case studies of befriending programmes in Northern Ireland were studied, reflecting variation in contextual variables, service user and provider characteristics. Data was collected via service documentation and semi-structured interviews (n = 46) with stakeholders involved in the delivery and receipt of befriending interventions.

### Results

Eight initial programme theories were generated, which were 'tested' in the case study analysis to uncover context-mechanism-outcome relationships. Mechanisms identified included reciprocity, empathy, autonomy, and privacy which were triggered in different contexts to support the alleviation of loneliness and social isolation. Reciprocity was 'triggered' in contexts where service users and befrienders shared characteristics, the befriender was a volunteer and befriending took the form of physical companionship. Contexts characterised in terms of shared experiences between befriender and service user triggered empathy. Autonomy was triggered in contexts where befriending relationships were delivered long-term and did not focus on a pre-defined set of priorities. Privacy was triggered in contexts where service users had a cognitive/sensory impairment and received one-to-one delivery.

**Data Availability Statement:** All relevant data are within the manuscript and its Supporting Information files.

**Funding:** This study is part of a PhD project that was undertaken by the first author, OAF, supervised by NMC and MD. The PhD Studentship Award granted to OAF was funded by the Northern Ireland Government Department of Economics. The funder had no role in the study design, data collection and analysis, decision to publish, or preparation of the manuscript.

## Conclusion

This study improves understanding about how and why befriending interventions work. Findings indicate that services should be tailored to the needs of service users and take into consideration characteristics including mobility, impairments e.g. physical, sensory and/or cognitive, as well as the influence of service characteristics including payment for befrienders, fixed/long-term befriending relationship, one-to-one support and the impact of non-verbal communication via face-to-face delivery.

## Introduction

There is increasing awareness about the potential harm that loneliness and social isolation can cause [1]. Loneliness can be defined as an undesirable subjective experience, related to 'unfulfilled intimate and social needs' [2] whereas social isolation can be understood as an objective concept capturing the absence of relationships, contacts or ties with other people [3]. Although loneliness can occur at all ages, it is a particularly common problem among populations of older people [4] as the opportunities for social contact are limited by various factors such as death of peers, physical limitations such as sensory deficits that limit communication, and/or mobility limitations that restrict their ability to visit family and friends [5]. Loneliness and social isolation have been associated with a range of health conditions and studies have demonstrated how they can lead to adverse physical and mental health consequences, especially among the older population [6, 7].

Across the UK and internationally, a number of initiatives have been deployed by health and social care providers as well as community and voluntary organisations, to address the issue of loneliness and social isolation. Befriending is a popular type of initiative that is mainly implemented to provide relationships and social contact to individuals experiencing loneliness and isolation in community and residential settings [8]. These services are now part of the social landscape in several countries especially the UK, US, Canada, Australia and Europe [9]; and several function at a grassroots level with aims to fill the emotional and social gap that is not being met by existing statutory and social service provision [10]. While befriending services are offered to a diverse range of populations of all ages and needs [11], this study focuses on support for older people. There are various definitions of the age range of 'older' populations [12] however most services broadly define older individuals as aged "50+ or 60+" p.17 [13]. There is no reliable estimate of current befriending provision but Mulvihill [11] identified in 2011 that there were over 3500 different befriending schemes for older people in the UK. Befriending services have also been identified as a common type of intervention delivered in Northern Ireland (NI) to reduce feelings of loneliness among older people. In NI, there are several networks which support befriending services, such as the Befriending Network [14] which was launched in 2011 and currently encompasses 26 organisations delivering befriending across NI. Like other parts of the UK, the increased provision of befriending services is part of the strategy [15, 16] to reduce loneliness and isolation among older people in NI.

Despite the increasing popularity of these interventions, several reports have highlighted that befriending remains poorly understood, under-researched and under-resourced [17–20]. Previous research on befriending has predominantly been outcome-focused with aims to assess its effectiveness for improving psychological health outcomes using randomised controlled trials (RCTs) [21–24]. In a systematic review of RCTs and quasi-experimental trials of

befriending interventions [17], evidence of an overall improvement benefit in patient-reported primary outcomes among people with physical and mental health conditions was reported, but the effect size was small and not statistically significant. Authors concluded that the current evidence base does not allow for firm conclusions on the effects of befriending on specific outcomes [17]. Mead, et al. [19] conducted a systematic review to identify the clinical effectiveness of befriending as a social intervention, particularly focusing on individuals who were experiencing depressive symptoms or emotional distress. Similar to existing literature [25], findings highlighted that befriending had a 'modest but significant effect' on depressive symptoms in the short-term [19].

A weakness of the experimental format, particularly relevant to the evaluation of befriending services, is the 'Martinson problem' [26], whereby inconsistent findings have been produced on the effectiveness of befriending because of neglect of contextual variation brought about by aggregate-level (between group) analyses, and failure to embrace the complexity of befriending services or address mechanisms of change [17]. Another limitation of the experimental format is that it does not place emphasis on how or why programs work or fail [27].

On the other hand, with the exception of Lester, et al. [9], previous qualitative research on befriending has focused predominantly on identifying the experiences of the befrienders and/ or service users [28–33]. Studies have identified common challenges such as the commitment required and benefits including the development of genuine relationships, companionship [32], mobility and engagement in various activities [33]. Qualitative research has not sought to identify or provide information on the mechanisms that produce the outcomes observed in befriending interventions such as the alleviation of loneliness and/or social isolation.

There is limited explanation in existing literature of the theories underpinning befriending interventions [34, 35] including a lack of understanding of how and why befriending interventions function differently for different people in different settings, or which populations they are most suited for [18]. Despite this, there is increasing investment in befriending interventions and policy-makers are committed to increasing the availability of this type of intervention to reduce loneliness and isolation among older people. Age UK, a major provider of befriending services in the UK [36] have published their concerns that service providers are experiencing a high demand to provide initiatives to combat loneliness, even with the lack of consensus about their effectiveness.

Befriending interventions are highly contextual as they are introduced within complex social systems, comprised of an interplay of individual, interpersonal and institutional characteristics, and the wider infrastructural system (see Table 1 below).

All befriending interventions are conditioned by the action of layer upon layer of contextual influences, hence are in constant transformation. Therefore, evaluation of such interventions need to consider the settings (context) within which it is implemented. Contextual contingencies are likely to influence implementation success and failure, how the intervention achieves impact, why their impacts vary and also the extent to which befriending interventions can be successfully transferred from one context to another [37]. In this research, a realist evaluation, underpinned by the philosophy of scientific realism, was utilised to identify the intricate relationships and underlying processes of these interventions. Using five case studies of befriending interventions, the study addresses some of the limitations of previous literature summarised above by focusing on contextual variation and the identification and action of mechanisms.

## Research aim

This study aimed to address the gap in the evidence-base by going beyond the identification of 'what works' to gain an in-depth understanding of how befriending interventions work, for whom, and in what circumstances. The overarching research question is:

**Table 1. Contextual layers of reality [26] in regard to befriending interventions.**

| Contextual layers | Description |
|---|---|
| Individual capacities of the key actors | This takes into consideration whether befrienders have the appropriate motivations, capabilities, experience and expertise to deliver an effective service that will alleviate feelings of loneliness and social isolation in the older person. It also takes into consideration the service users' motivations, capabilities, health and mobility. |
| Interpersonal relationships supporting the intervention | This considers if the lines of communication between the service managers, coordinators and befrienders, are supportive or damaging to the delivery of the service to service users. |
| Institutional setting | This considers if the ethos, culture and characteristics of the befriending intervention supports the alleviation of loneliness and social isolation among its service users. |
| The wider infra-structural system | This considers if there are policies that support the implementation of befriending interventions; if there are resources to support the older person to remain active and engaged in their community; and if a neighbourhood-level response exists which improves the community's own capacity to tackle loneliness. |

Table 1 shows the contextual layers applicable to befriending interventions. It considers the: a) individual capacity of the key actors including befrienders and service users, b) interpersonal relationships between the service managers, coordinators and befrienders that support the intervention, c) institutional settings such as the ethos, culture and characteristics of befriending interventions, and d) wider infra-structural system such as policies that support the implementation of befriending interventions, as well as the availability of resources and capacity to support older people.

What contextual factors and mechanisms are necessary for befriending interventions to produce positive outcomes such as an alleviation in loneliness and social isolation among the older population?

## Methods

### Evaluation design

This study employed a realist evaluation research methodology and was conducted in three phases as recommended by Pawson and Tilley [26] (Fig 1).

### Phase one: Development of initial programme theories (IPTs)

IPTs are hypotheses that explain how and why a programme works (or does not work) within a particular context. These hypotheses are then tested using the best available evidence to confirm, refute or refine the theory. A number of approaches were used to elicit the IPTs in line with the strategies advised by the RAMESES guidelines [38, 39]. These included:

1. Findings from a previous profile study [40]–this study utilised a modified version of the TIDieR framework [41, 42] to structure an interview schedule to identify the types of interventions delivered in NI to alleviate loneliness and/or social isolation among the older population, and to elicit the views of service providers about mechanisms perceived to be associated with reducing loneliness.

2. Literature review–tacit theories were extracted from the literature (i.e. befriending service guidelines, reports, evaluations, and relevant quantitative and qualitative research) about features that make befriending interventions work. Following realist principles [43], abstract theories at the middle-range level were harnessed to guide the development of the initial programme theories by highlighting key concepts that might be influential.

**Development of IPTs informed by:**

- Interviews with service personnel of loneliness interventions
- Review of programme-related documents (guidelines, reviews, research studies, discussion papers)
- Informal discussions with programme developers and service personnel

**Synthesis across cases**

- Case-specific CMO configurations are collated across cases and reviewed in order to identify demi-regularities
- Translation of demi-regularities and programme theories to identify middle-range theories

1. Development of initial programme theories (IPTs)

2a. Testing of IPTs using empirical data (Data collection)

3. Synthesis of programme theories

2b. Testing of IPTs using empirical data (data analysis)

**Testing of IPTs using empirical data collection (Data collection)**

**Study Design:** Qualitative multiple case study design

**Data collection:** Iterative process involving semi-structured interviews with stakeholders who have been involved in the planning, development and implementation of befriending services (i.e. service managers/coordinators, befrienders, service users, family members of service users), as well as review of relevant service documents

**Testing of IPTs within cases using empirical data (Data analysis)**

- Relationships between C,M,O assessed
- Outcome patterns observed are examined to identify context-mechanism-outcome configurations
- Refinement of the initial programme theories according to the CMO configurations and emerging patterns from the study

**Fig 1. Realist evaluation process and phases according to Pawson and Tilley as applied to study of befriending interventions.** This figure shows the three phases according to the framework for realist evaluation outlined by Pawson and Tilley [26], including: 1) development of the IPTs, 2) testing of IPTs (data collection and analysis), and 3) synthesis of IPTs across all cases.

3. Stakeholder feedback–informal discussions and meetings were held with stakeholders involved in the planning, development and implementation of befriending services, e.g. managers and coordinators of befriending services. The purpose of these discussions was to identify: 1) factors that influenced the implementation of befriending services, 2) initial observations about how and for whom befriending services work and why, and 3) objectives of the realist evaluation; all of which produced information that was used to inform the programme theories.

Results from these activities highlighted the heterogeneity of befriending services. Key components were identified from the sources above which were considered important in understanding the architecture of befriending services. These characteristics formed a conceptual framework from which IPTs were generated and organised (see S1 Appendix in S1 File). The IPTs identified are framed as 'if-then' statements [44] and include the context (C), mechanism (M–resource, M–reaction) and outcome (O) propositions to explain how befriending services work.

## Phase two: Data collection and analysis

**Research setting.**   A multiple case study design was utilised with purposive sampling to select existing befriending services operating in NI as 'cases'. Befriending services were purposively sampled to maximise the opportunity for accessing a diverse range of contexts, and were identified via the 'Befriending Network NI', a directory of 26 member organisations published by 'Volunteer Now' in collaboration with the Befriending Network [8]. To be eligible, services had to:

- Have an explicit goal (primary or secondary goal) of preventing or reducing loneliness and/or social isolation.

- Be currently provided to older people in NI.

- Be delivered in-person.

- Have carried out health and safety risk assessments of service user's home; and

- Have access to service activity data such as evaluation reports.

To be eligible, it was not required that services specified the age-range of its service users, only that they identified themselves as offering the service to older people. Following this, five cases were identified with varying characteristics (S2 Appendix in S1 File).

## Data collection

Consistent with the realist evaluation logic of inquiry, a combination of methods was used, including realist semi-structured interviews with representatives of several stakeholder groups and review of service documentation for each case. The selection of participants was based on their 'CMO investigation potential' [26], hence, within each case, the researcher sought to obtain the perspective of service managers, coordinators, befrienders, service users and family members/next of kin of service users aged over 16 years. Participants (service users, befrienders and family members of service users) were eligible for this study if they had been involved

in the befriending service for a minimum of three months, and were excluded if they were unable to participate in an interview because of severe cognitive impairment or significant communication difficulties. For service users with dementia, capacity to participate in an interview was assessed by a person with a duty of care, consistent with the Health Research Authority guidelines [45].

### Ethical approval

Ethical approval was granted by Queen's University Belfast, the School of Medicine, Dentistry and Biomedical Science Research Ethics Committee prior to commencing data collection (Ref: 19.23).

### Recruitment of participants and data collection

The researcher initially contacted the service manager/coordinator to provide an explanation of the research and seek approval for the research to be conducted within their service. The service manager then identified eligible participants from each of the groups (e.g. befrienders, service users and their family members/next of kin), and distributed the relevant study materials (participant information sheets and consent forms) to them. Eligible participants were asked to indicate their consent to be contacted directly by the researcher, who then made contact with these individuals to arrange an interview date, time and location.

Semi-structured interviews were conducted using an interview guide tailored for the specific participant group (see S3-S6 Appendices in S1 File). The interview schedules explicitly discussed the IPTs in a question format, to either generate programme theories or interrogate the programme theories already identified by giving participants the opportunity to confirm, refute or refine the theory. The 'teacher-learner' approach was employed whereby the researcher taught the interview participant the programme theory and the interview participant, in turn, taught the researcher about the components of the theory as they happen in practice [26]. All interviews were conducted in person by the lead author (OAF), audio-recorded and transcribed verbatim. Recruitment of participants and interviews were carried out until consistent patterns were observed in the analysis which demonstrated that data saturation had been reached, e.g. no new information emerged from the interviews. Review of relevant documents (e.g. evaluation reports and newsletters) was also carried out as a means of triangulation and served two purposes: 1) to provide a broad sense of the context of each befriending organisation, and 2) to identify components of the CMO configuration that supported or refuted findings from the interviews.

### Data analysis

The data analysis utilised a retroductive approach [46], by applying both inductive and deductive analytical processes to multiple data sources (e.g. interview transcripts and service documents) while also incorporating the researcher's own understanding to uncover generative causation. This process required the researcher to move back and forth between the IPTs and the data, to identify elements of contexts and mechanisms that explained the outcomes, and to refine the IPTs according to the CMO configurations and emerging patterns. The semi-structured interviews were the most heavily featured as they were the only data source that contained extractable CMO configurations in their entirety. These interviews were used as the starting point and then the researcher moved on to service documents to triangulate and inform the testing and revision of the theories by identifying information that would help to support/refute/refine the CMO configurations.

The interviews were transcribed verbatim and were imported with relevant service documents into a computer-assisted qualitative data analysis software (NVivo 11) to allow for coding of the data following the guideline outlined by Gilmore, et al. [47]. The coding process involved examining the data sources and extracting information on contexts, mechanisms and outcomes from sections of the text that provided supporting evidence for the particular IPT under analysis. Data from each case was initially analysed separately to enable comparison and synthesis of findings across all cases.

Following this, the CMO configurations (CMOCs) pertaining to the refined programme theories across cases were collated and patterns in the data (demi-regularities) were searched for within the combined programme theories to identify similarities in contexts, mechanisms and outcomes. Once demi-regularities were identified, the literature was searched for substantive theories to enhance the explanatory endeavour of the study, and to support the shift from case-specific programme theories to middle-range theories. These theories are specific enough to clearly explain the phenomenon but general enough to be applied across cases of the same type. This process adhered to the guidance by Gilmore, et al. [47] and the RAMESES II guidelines for realist evaluation [38].

## Rigour

A number of strategies were used to address quality and rigour of the research. For example, two methods of data collection were used to ensure triangulation, and a clear audit trail was established through audio recordings, emails and interview notes. Service documents obtained were reviewed to identify relevant information to inform and support findings from the interviews. Moreover, recruitment of participants and interviews were carried out until consistent patterns were observed in the analysis which demonstrated that data saturation had been reached. Furthermore, the researcher coded initially the first three transcripts and the codes derived were reviewed by the primary supervisor (NMC) and discussed to develop joint understanding of what constitutes a context, mechanism and outcome in befriending interventions. Subsequent transcripts were coded independently by the researcher and both supervisors of the research project (NMC and MD) reviewed and oversaw the data collection and analysis process.

## Phase three: Synthesis

## Results

A total of 46 interviews were conducted between July 2019 –January 2020 across five befriending services: 18 participants from case A, 12 participants from case B, 11 participants from case C, two participants from case D, and three participants from case E. The total sample (n = 37 female, n = 9 male) across all five cases comprised of: service managers/coordinators (2 service managers and 4 service coordinators), befrienders (n = 17), service users (n = 14), and family members related to the service users interviewed (n = 9). Service users were either widowed (n = 7), married (n = 3) or were single (n = 4). Nine service users lived alone and the remaining five lived with family members. Initial contact with the befriending service was often initiated by a relative or professional, often following spousal bereavement or deterioration in health which caused the individual to have reduced mobility and increased dependency. Four service users completed the interview alone whereas the remaining 10 had a relative present.

The five cases generated a total of 40 PTs (7–11 per case). The PTs from each individual case study were summarised to produce a concise description of how befriending services work, while also preserving the insights gained from these varying service contexts. This was

presented to stakeholders during two meetings, when the implications of these programme theories for practice were discussed. This was an important step to consolidate the findings and to produce guidance for future practice and research in different contexts. The fully synthesised PTs are displayed in S7 Appendix in S1 File. Synthesis of these CMOCs across cases produced four demi-regularities. Findings have been arranged according to these configurations and the most prominent quotes from participants are presented to illustrate each CMOC.

## Demi-regularities

*Reciprocity* enhances the longevity of the relationship and the development of a friendship, and is triggered when befrienders fulfil the role voluntarily and deliver the service through face-to-face in-person communication, particularly in contexts where service users have mobility or sensory problems. Similarities between befriender and service user help to promote reciprocity early in the relationship.

Similarities between the characteristics and preferences of befrienders and service users is particularly helpful during the early phases of the befriending relationship as it increases the propensity for bonding and attachment, and influences the establishment of reciprocity (a mutually beneficial relationship). Previous research [9] and findings from this realist evaluation study highlight that reciprocity is likely to occur when the befriender and service user belong to the same generation, share a common culture and social background, have common interests, and/or are based in the same location. The principle of homophily [48], proximity [49] and Hurley's decision to trust theory [50] explain this pervasive pattern of friendship choice. A high compatibility between the befriender and service user facilitates the development of trust and this is supported by sub-components within Hurley's 'decision to trust' theory [50]. The theory highlights seven situational factors that impact on the development of trust: 1) security, 2) number of similarities, 3) alignment of interests, 4) benevolent concerns, 5) capability, 6) predictability and integrity, and 7) level of communication. According to Hurley [50], individuals tend to easily trust others who share similar interests and appear similar to themselves. When an individual is deciding how much to trust someone, they often begin by tallying up their similarities and differences, hence it is more difficult to trust people who appear more different [50]. However, differences in personalities and interests can also be complementary to the relationship and aid in conversation-making [51].

> **Quotation:** *"I consider my service user as my friend. . .If I left the befriending service, I would still go and visit her as she is housebound. We've connected so well [though] we are different. I'm quite shy and quiet and she is very chatty so she can talk away and I'd just listen and this works for us. So we have different personalities but it works. . . I wouldn't say that you have to have exactly the same interests because sometimes having different interests makes it more interesting and gives you things to talk about. They do say that opposites attract."* Interview BRS26B –Befriender"

**C**: Service user is housebound. Befriender and service user have contrasting interests and personalities.

**M**: **Resource:** Difference in characteristics complements each other.

**O**: Different interests acts as a conversation stimulant. Good connection is developed between the befriender and service user.

The homophily principle is the tendency for individuals to bond with others similar to themselves [48]. Similar personalities, likes and dislikes facilitates engagement in similar

activities thus, matched individuals are likely to mutually reinforce each other's behaviour patterns [51, 52]. In contexts where service users had a cognitive impairment, e.g. dementia, matching that prioritised opportunities for them to continue with existing hobbies and interests was important as it allowed them to continue engaging in activities that had become routine and therefore helped them to sustain their personal identity and sense of autonomy. Interviews with stakeholders confirmed this as seen in the quotation below.

> **Quotation:** *"I started using the befriending service to keep my mind active otherwise it will go down and I will feel terrible. . .I love board games and wanted for someone to come and play them with me so they brought a befriender who likes board games. We have good banter and I really enjoy his company. He was the perfect match for me."*–Interview BRS1A –Service user

**C:** Service user has dementia
**M: Resource:** Befriender engages in activity of interest to service user, e.g. board games, which provides mental stimulation and keeps the service user's mind active.
**O:** Service user enjoys the company of the befriender and a reciprocal relationship is established.

The proximity principle is the tendency for individuals to develop interpersonal relationships with those who live close by [49], hence the advantage of befrienders and service users being in the same geographical location. This was first documented by Newcomb [49] who found in his study of the acquaintance process that people who interact and live close to each other are more likely to develop a relationship. Reciprocity was also more likely to be developed when befrienders were unpaid because service users recognised that they were choosing to visit them rather than being under any financial obligation to do so. Moreover, befrienders who engaged in the service as volunteers found the role satisfying as it enabled them to fulfil a sense of duty in giving back to the community and making a positive contribution to the lives of service users.

> **Quotation:** *"Doing this voluntarily, she is more laid back and it is more of a friendship. It doesn't feel like a job for me. The ethos of befriending is that you are giving up your time. You don't seek money for it because you want to do it and you get satisfaction from being able to help someone else. My service user and I have established a deep bond and I believe that we are connected on so many levels, such as emotionally. This would probably not have happened if I was getting paid. It would have probably been the same type of interaction that would occur if they had a carer. So it would be a very casual type of interaction that might not deepen and we'd remain acquaintances rather than friends."* Interview BRS21A - Befriender

**C:** Befriender has a pre-existing desire to want to voluntarily give back to the community.
**M: Reasoning:** Befriender feels satisfied at being able to help service user.
**O:** A friendship is established between the befriender and service user and both parties are connected emotionally.

Reciprocity was facilitated through face-to-face in-person communication whereby befrienders were able to see the positive impact that they were making to the life of their service user which reinforced their drive and motivation to continue delivering the service. Additionally, this interaction method allowed for the mutual exchange of refreshments (e.g. offering a cup of tea) which confirmed the 'everydayness/normalcy' of the relationship. Moreover, in circumstances where service users had reasonable/good mobility, the in-person visits broke the monotony of being at home as the befriender created or facilitated external social activities and provided the support/companionship needed for the service user to leave their home and engage in activities in their community, thus potentially expanding their social network.

**Quotation:** "*I like to be out and seeing people and meeting new people. I feel like part of my community again and it's a nice, reassuring feeling. I was isolated before. I don't like talking on the phone as you are just sitting in the house. You don't know the person as you can't visualise what the person looks like... It helps me to trust my befriender. If I didn't trust them, I would always be on edge and won't be relaxed and myself around them.*" BRS3A –Service user

**C:** Service user was isolated but has good mobility.

**M: Resource:** Befriender provides physical contact and opportunities for service user to engage in their community.

**Reasoning:** Service user finds it easier to build a relationship with befriender. Service user feels relaxed around befriender and trusts them.

**O:** Service user is integrated into their community.

*Empathy is important for establishing an understanding relationship and is triggered* when befrienders have similar experiences to the service user. Empathy is promoted via non-verbal communication and facilitated by in-person service delivery.

Empathy can be described as an affective response which recognises and attempts to understand the suffering of an individual through emotional resonance [53]. Non-verbal communication such as eye-contact, facial expressions, open body posture and touch are primary vehicles for expressing emotions [53]. Hence, face-to-face communication between the befriender and service user was found to promote the development of empathy.

**Quotation:** "*The face-to-face allows you to be more empathetic to their situation and the difficulty that they are going through [with their dementia] and you can read their body language. You can see their facial expressions and their reactions to what you are saying or doing. You can see how it is affecting them. The face-to-face helps to develop a deeper connection than you would get over the phone.*" BRS17A - Befriender

**C:** Service user has dementia and is going through difficulties.

**M: Resource:** Befriender can see the non-verbal cues such as facial expressions, reactions and body language, and better understands the service user's experience. Befriender can display empathy to service user.

**O:** Connection between befriender and service user is strengthened.

The ability to empathise was enhanced when befrienders had similar personal experiences to the service user, particularly with respect to specific health conditions (such as arthritis). Such befrienders were likely to be able to recognise and understand the feelings of the service user and thus deliver an attuned response. Moreover, service users valued shared experiences with their befriender as it was reassuring and affirming of their situation and they perceived their befriender to have a true understanding of what they were going through which facilitated stronger and trusting bonds between both parties.

**Quotation:** "*The volunteers within this organisation suffer a long-term condition themselves because how can you offer empathy to someone living in pain if you don't have pain yourself? It helps you to connect more because you can relate with them. You are living with a long-term condition so you know what it is like... I think empathy is the word. Because I know what she is going through and what it is that she needs help with which is very reassuring for her. This makes me more understanding and patient.*" Interview BRS30D - Befriender

**C:** Befriender suffers from a long-term health condition similar to service user.

**M: Resource:** Befriender has good knowledge and understanding of the difficulties of living with and managing a long-term health condition therefore is better equipped to provide appropriate support and is more empathetic.

**Reasoning:** Befriender connects more with service user and is able to relate better with them. Befriender is more patient and service user feels reassured.

**O:** An empathetic and understanding relationship is developed between the befriender and service user that is meaningful and reciprocal in nature.

In circumstances where befrienders do not have personal experience of the situation, e.g. health condition of the service user, it is assumed that they would require more effort to develop empathy [20]. In these circumstances, training was beneficial as it allowed befrienders to develop knowledge and better understanding of the illness of the service user and therefore feel better prepared/equipped to deliver the service. Training was also valuable in circumstances where befrienders had personal experience or previous knowledge as it provided new insights about the illness and awareness of the different experiences of the illness.

> **Quotation:** "*The befriending service trained me before I started delivering this service as a befriender. I felt confident as I was adequately trained. Even though I also have a similar health condition as my service user, I found the training beneficial as it really prepares you for what could potentially happen. Everyone is different and not everyone will react in the same way or experience the condition in the same way as you as it's important to be aware of this and know of how to deal with it which the training helps you with.*"–Interview BRS30D - Befriender

**C:** Befriender has similar health condition as service user however, received training prior to service delivery.

**M: Resource:** Training provided insight into the different experiences of the illness and how people can respond differently.

**Reasoning:** Befriender feels adequately prepared.

**O:** Befriender is more equipped and confident in delivering the service.

*Autonomy* enhances the normalcy of the befriending relationship and is triggered by providing service users a flexible befriending relationship that is long-term in nature, and is particularly beneficial to service users with reduced mobility.

Service users that received a long-term befriending relationship were likely to feel more autonomy in the relationship as they were provided the leisure of doing things that they found meaningful at their own pace. This enhanced the normalcy of the relationship as the activities were not restricted to a pre-defined set of priorities that is common with fixed-term relationships, thus could be led by the service user's needs and preferences which befrienders were able to respond to in a holistic manner. The long-term nature of the relationship was particularly beneficial for service users of older age and/or with reduced mobility who faced greater obstacles in social involvement, causing them to become housebound and experience a social network decay. Additionally, in circumstances where service users had little or no family support, the befriender played an active role as a link to the outside world so that service users felt engaged in their community. This was evidenced in interviews with service users in this study.

> **Quotation:** "*. . .My circumstances haven't changed. I still live alone and have reduced mobility and visual impairment. If my befriender stopped visiting then I definitely would start to feel lonely and isolated once again because the only difference in my life now than before I started receiving this service is that I now have someone coming to visit me on a set day and time every 2 weeks. It has become an important social routine in my life. . . My health*

*situation is still the same so I'm not able to leave my home and socialise for myself."* Interview BRS14E - Service user

**C:** Service user lives alone, has reduced mobility and visual impairment.
**M: Resource:** Befriender provides companionship and a social routine.
**O:** Service user would feel lonely and socially isolated should the befriending visits come to an end as their health condition hinders them from being independent and socialising in their community.

Carstensen's socioemotional selectivity theory [54] posits that older people attach greater importance to meaningful relationships as they age. Relationships that are continuous with the same befriender, are likely to be more meaningful as familiarity and trust are likely to be developed which ultimately strengthens the connection between both parties. Moreover, having the same befriender visiting was particularly important for service users with a cognitive impairment (e.g. dementia), a sensory impairment (e.g. sight loss), or a learning disability, as these individuals are generally more sensitive to changes in their environment and therefore more vulnerable to confusion and anxiety if different befrienders were to visit. In these circumstances, the befriender's role tended to focus more on the establishment of companionship and less on expanding the social network of these individuals.

**Quotation:** *"I think that it makes a difference that the same befriender visits the service user especially in this service as some have a learning or sensory disability and having different people would cause confusion. If a different person was to come to someone's house every week, the service user would get very mixed up with things because I provide a whole different attitude, outlook and behaviour and the next befriender could be the complete opposite. So this would confuse the service user. It would be hard to establish anything meaningful as there would be no time to. It's good to have that consistency. I also know how best to communicate with them specifically as I have learnt and adapted this over time. So I understand them better."* Interview BRS29C –Befriender

**C:** Service users have a learning or sensory disability.
**M: Resource:** Consistency in befriending relationship via same befriender visiting.
**Reasoning:** Service user does not feel confused.
**O:** A meaningful relationship is established between befriender and service user. Befriender has learnt the communication style of the service user and thus is able to communicate effectively with them.

In contrast, fixed-term befriending relationships are more rigid in their approach to support and tend to be goal-oriented and heavily task-focused [33, 34]. Service users who are more physically mobile are likely to benefit more from fixed-term befriending relationships as during the relationship, the befriender can engage in a range of task-based activities to boost the confidence and independence of these individuals so that they would develop the skills necessary to be able to cope without the support of the befriender. This realist evaluation identified that the fixed-term befriending relationship was beneficial during the period of service delivery as the service user reported developing a good relationship with the befriender, however, there was a resurgence in feelings of loneliness once the befriending sessions had come to an end.

**Quotation:** *"I only had eight weeks with my befriender. I would have liked for it to be longer because I really got used to her company. . . She really was like my personal friend. I was really sad when the service ended. Although I felt less lonely when she was visiting me, when this*

*stopped, I started to feel lonely again. I missed having that company around. I missed having someone actually spending quality time with me and being able to confide in her."* Interview BRS13D - Service user

**C:** Service user was lonely and received a fixed-term befriending relationship.
**M: Resource:** Befriender provided companionship.
**Reasoning:** Service user felt sad when the befriending relationship ended as a friendship had already been established.
**O:** Feelings of loneliness resurfaced when the befriending relationship came to an end.

Resurgence in feelings of loneliness likely occurred because the social circumstances of the service user remained unchanged after being discharged from the service. The objective of the befriending visits was to help them manage and cope better with their health condition and not to build the social connections of the service user (beyond the befriending relationship). Hence, long-term alleviation of loneliness is unlikely with a fixed term service model unless service users are enabled, with support from the befriender, to form friendships in their communities.

*Privacy* is important in developing a meaningful relationship, particularly for service users with a cognitive or sensory impairment, and is triggered when environmental distraction is reduced thus enhancing cognitive ability.

The interactions between the befriender and service user were mainly one-to-one which facilitated increased engagement and privacy, and eliminated external distractions more likely to be present in group interactions. This was particularly important for service users with a sensory or cognitive impairment. Service users with cognitive or sensory impairments described how they were likely to disengage in group environments, that the high stimulation could be intimidating, and could reinforce marginalisation and compound feelings of loneliness.

> **Quotation:** *"He gets lost in a group because of his Alzheimer's. He can't follow up on a lot of conversations. Because of his illness, he would not do well in a group. Also, because of his hearing, he can't have loud noises at all. His hearing is also not that good either so would find groups to be too noisy. There would be too many stimulations in a group and it would be too overwhelming for him. He is not comfortable in the midst of a lot of people. He would have to have one-to-one. . .to be able to know his befriender properly, trust them and develop a relationship with them."* Interview BRS38A - Wife of service user

**C:** Service user has Alzheimer's and hearing difficulties.
**M: Resource:** One-to-one format reduces unhelpful stimulations.
**Reasoning:** Service user does not feel as overwhelmed as they would in a group setting and can cope better.
**O:** A good connection is established via one-to-one interaction and trust is developed.

One-to-one interaction provides less distractions and makes the environment facilitative [55]. Moreover, service users felt more comfortable disclosing personal information in an environment that was perceived to be safe and conducive to the development of trust.

> **Quotation:** *". . .But with one-to-one, I'm the centre of attention. My befriender focuses solely on me and it makes me feel listened to and understood. I would rather my befriender comes to me and it's just the two of us. You can talk one-to-one but in a group, you might not want everybody to know what you are talking about. I'd feel more comfortable saying personal things to my befriender. . .It helps to build trust."* Interview BRS14E –Service user

**C:** Service user is visually impaired.

**M: Resource:** Befriender gives full attention and a significant amount of time to service user. One-to-one provides privacy.

**Reasoning:** Service user feels listened to and understood. Service user feels more comfortable disclosing personal information.

**O:** A trusting relationship is developed between befriender and service user.

In some circumstances, if the group activity is targeted at individuals with a specific health condition or experience, this can provide respite from focusing on their perceived differences from others as it creates an environment where the experience of the service user is normalised and allows them to interact with other individuals in the same or similar situation as them. Research shows that coming together in adverse circumstances can create a sense of camaraderie which is ultimately very supportive [56].

**Quotation:** *"There have been cases where service users [who have a learning, physical or sensory disability] have been brought together at events such as Christmas parties and there is an understanding that they are not alone and that there are many people [going through a similar experience] who receive the same or similar service to them. So they see that there is a commonality and that it is quite a normal thing to be happening. From this, friendships have developed from these events."* Interview BRS35C –Service Coordinator

**C:** Service user has a disability.

**M: Resource:** Group interaction provides a normative environment.

**Reasoning:** Service user feels reassured of their situation.

**O:** Friendships are developed.

However, in circumstances where individuals have a chronic degenerative condition, groups can expose service users to others who have advanced or more progressive illness thus causing fear and/or anxiety. In general, group interaction may be suitable for individuals who are more physically mobile and can benefit from the provision of opportunities to expand their social network and re-integrate in their community.

## Discussion

The objective of this study was to empirically understand how befriending services work to alleviate loneliness and social isolation among the older population. The study findings uncovered four demi-regularities, highlighting key mechanisms by which befriending services work to produce outcomes such as alleviation in loneliness and social isolation: 1) reciprocity, 2) empathy, 3) autonomy and 4) privacy. Reciprocity echoed the principle of homophily [48], proximity [49] and Hurley's decision to trust theory [50] and drew attention to different strategies such as careful matching of befriender and service user, utilisation of volunteer befrienders, and in-person visits. Empathy was elicited through face-to-face service delivery and shared experiences between the befriender and service user (e.g. illness or bereavement). Autonomy was triggered via long-term support with the same befriender and echoed principles of Carstensen's socioemotional selectivity theory [54]. Privacy was facilitated via one-to-one interaction between the befriender and service user and was particularly beneficial for service users with a sensory or cognitive impairment. The health status of service users is a contextual condition that can influence service delivery, particularly activities undertaken with their befriender. For example, service users who are frail and have reduced mobility, or have a cognitive impairment such as dementia, are less likely to be reintegrated into their community due to the degenerative process of the illness and thus benefit more from long-term support

provided by a befriender. Hence, the goals of befriending services should be individualised. In some contexts, these goals may not prioritise the expansion of the social network of the service user beyond the befriending relationship, or the empowerment of service users to engage in activities in their community without the support of a befriender. It is important for befriending services to be explicit about their goals; to ensure that these goals are tailored to the needs of service users; and that activities undertaken are relevant to the achievement of these goals. Whilst findings suggest that the provision of long-term support should be prioritised for service users with cognitive, sensory or physical impairments, services should also be mindful that some individuals with these conditions wish to be further integrated into community life. In circumstances where fixed-term support is provided, services should aim to increase the independence of service users and expand their social network beyond the befriending relationship. However, when the relationship comes to an end, a needs assessment should identify service users who need to be linked to a longer-term intervention.

Matching befrienders and service users based on similar socio-demographic characteristics was found to influence the success and longevity of the relationship. This was particularly important when service users had a cognitive (e.g. Alzheimer's or dementia) or sensory condition (e.g. sight and/or hearing loss). In these circumstances, matching based on shared interests/hobbies allowed for service users to continue to engage in activities that sustained their personal identity. Where service users had a chronic physical health condition, e.g. arthritis, matching with a befriender who had the same/similar illness created a bond where service users felt better understood because befrienders could relate to and empathise with the challenges of the illness. Additionally, in circumstances where service users had a learning disability and experienced communication difficulties, consideration of the skills-set of befrienders during the matching process was important as it influenced the capacity of the befriender to cope and to deliver a more effective service. These characteristics should be considered within the matching process and needs assessments.

The voluntary nature of the befriending role was integral to the establishment of reciprocity and friendship building. However, services find it difficult to recruit enough volunteers to satisfy demand. Findings suggested that paying befrienders is not necessarily a solution to this problem because a paid service model can threaten the establishment and maintenance of the friendship. Hence, services need to explore other means of meeting the demand for befrienders, such as methods of utilising the existing pool of befrienders more efficiently. For example, services might actively seek to identify clients who could be supported and empowered to make connections within their communities, therefore relying less on their befriender. Support from the befriender could be gradually withdrawn, or alternative means (such as telephone or virtual support) used to maintain a connection. As a result, befrienders may be in a position to support several service users simultaneously, or to move on to another client after a shorter period of time. The preferences and motivations of befrienders also need to be considered in these circumstances.

The revised programme theories (S7 Appendix in S1 File) explain how these four mechanisms (reciprocity, empathy, autonomy and privacy) were triggered in particular contexts to produce the outcomes observed. Within these theories, different contextual layers are represented. For example, regarding the individual capacity of key actors such as service users and befrienders, findings from this study highlighted how the health, mobility, motivation and capabilities of these individuals can impact the success of the befriending relationship. In regard to interpersonal relationships, befrienders who felt supported and received appropriate training and supervision from service managers reported benefits to their relationship with service users. With regards to the institutional setting, characteristics such as monetisation of befrienders and the short-term/long-term nature of the befriending relationship impacted

outcomes. Future research should examine the influence of the wider-infrastructural system. Findings from this research highlighted that geographical proximity between the befriender and service user could influence the establishment of a relationship between both parties.

Limitations of this realist evaluation include that in a few cases, some stakeholder groups (e.g. service users and befrienders) were under-represented. Where service users were under-represented, there may have been less data available about the outcomes of the service, given that users are generally more sensitized about the programme outcomes. However, when data is combined across cases, there is ample information about the processes and outcomes of the services. Future research could widen the sample of organisations and include previous service users and those with failed befriending relationships in order to identify contexts where befriending is likely to be unsuitable. In addition to this, previous service users can inform questions about the long-term impact of befriending services.

The current COVID-19 pandemic presents a risk factor for increased loneliness and social isolation due to restrictions of movement and guidelines (i.e. stay-at-home restrictions, lockdowns and quarantines) enforcing physical distancing and social exclusion which has been implemented in many countries [57]. Meeting people in close proximity can now be a source of fear and anxiety, especially for service users from the older population who are more susceptible and at increased risk of severe impact from the virus [58]. During the pandemic, some befriending services have shifted to digital remote delivery (e.g. telephone or video consultations). Findings from this realist evaluation indicate that whilst this is likely to have less impact on relationships that are already established, remote delivery makes the establishment of new relationships more difficult. The lack of social cues may hinder the development of a deep connection in new befriending relationships, particularly among service users with a sensory or cognitive impairment and thus, the quality of these relationships may be threatened until in-person services are restored. Moreover, findings indicate that restrictions on face-to-face interaction may hinder the establishment of important mechanisms such as reciprocity and empathy.

## Conclusion

Research regarding loneliness is extensive and has gained increased attention over recent years. However, the older population are a heterogenous group and therefore studies targeting specific groups within this population are important. Existing knowledge is limited in terms of what loneliness interventions are most appropriate, for whom and how. Although befriending services are popular across the UK, there are gaps in current research on how and why these interventions function differently for different people in different settings, and which populations they are most suited for. This study is the first to employ a realist evaluation methodology to examine the impact of befriending services on the alleviation of loneliness and social isolation among the older population. Therefore, the findings contribute to new knowledge by identifying contextually relevant evidence for improving the implementation and effectiveness of befriending services. These findings may help policymakers make informed choices based on evidence about which type of service to use and how to adapt them to their local contexts. Moreover, practitioners are able to develop and improve the characteristics and targeting of future befriending services.

## Supporting information

**S1 File.**
(DOCX)

## Acknowledgments

The authors would like to thank the participants that took part in the study.

## Author Contributions

**Conceptualization:** Olujoke A. Fakoya, Noleen K. McCorry, Michael Donnelly.

**Data curation:** Olujoke A. Fakoya.

**Formal analysis:** Olujoke A. Fakoya, Noleen K. McCorry, Michael Donnelly.

**Investigation:** Olujoke A. Fakoya.

**Methodology:** Olujoke A. Fakoya.

**Project administration:** Olujoke A. Fakoya.

**Supervision:** Noleen K. McCorry, Michael Donnelly.

**Visualization:** Olujoke A. Fakoya, Noleen K. McCorry, Michael Donnelly.

**Writing – original draft:** Olujoke A. Fakoya.

**Writing – review & editing:** Olujoke A. Fakoya, Noleen K. McCorry, Michael Donnelly.

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
