## [Decision Letter · Decision Letter 0]

15 Jul 2021

PONE-D-21-14257

How do befriending interventions alleviate loneliness and social isolation among older people? A realist evaluation study

PLOS ONE

Dear Authors,

Thank you for submitting your manuscript to PLOS ONE. After careful consideration, we feel that it has merit but does not fully meet PLOS ONE’s publication criteria as it currently stands. Therefore, we invite you to submit a revised version of the manuscript that addresses the points raised during the review process.

We look forward to receiving your revised manuscript.

Kind regards,

Marcel Pikhart

Academic Editor

PLOS ONE

Journal Requirements:

Reviewers' comments:

Reviewer's Responses to Questions

**Comments to the Author**

1. Is the manuscript technically sound, and do the data support the conclusions?

Reviewer #1: Yes

Reviewer #2: Yes

2. Has the statistical analysis been performed appropriately and rigorously? 

Reviewer #1: Yes

Reviewer #2: N/A

3. Have the authors made all data underlying the findings in their manuscript fully available?

Reviewer #1: Yes

Reviewer #2: Yes

4. Is the manuscript presented in an intelligible fashion and written in standard English?

Reviewer #1: Yes

Reviewer #2: Yes

5. Review Comments to the Author

Reviewer #1: Befriending intervention has gained great attention in the field of health care. Specifying individual and contextual factors that are associated with positive effects of befriending intervention represents an important extension of previous studies. The goal of this research was to explore how befriending interventions work effectively to mitigate loneliness and social isolation among older people.

In this study, realist evaluation was conducted to clarify causal processes in different contexts of befriending interventions. The sample seemed suitable for the study aim. The author(s) did a good job of collecting and analyzing data. The author(s)' presentation of realist evaluation seems clear and reasonable. The findings of four mechanisms were theoretically significant and provide crucial implications for future befriending services. Considering these strengths, though, as I read the manuscript I found some areas in which I would have appreciated greater clarity. Below are more specific comments regarding various parts of the manuscript:

1. The introduction section felt somewhat short though the processes of realist evaluation needed to be elaborated in the Method and Results (i.e., more than 16 pages for Method and Result but only 3 pages for the Introduction). Although this suggestion is purely stylistic, I would expect to get enough background information in the Introduction section. Specifically, I was kind of confused about the term “older people”, and I found myself wondering about the definition of this population in the current study. Moreover, there was also limited information about “loneliness” and “isolation” which are keywords of this manuscript. Therefore, I would suggest that author(s) highlight the importance of alleviating “loneliness” and “isolation” in older people.

2. This study aimed to uncover contextual variation and underlying mechanisms of befriending services and address the gap between prior researches. However, it seems still unclear how your study extended past work in this area. It would be important to be more specific. For example, what are “inconsistent findings” on the effectiveness of befriending? What are the brief results of randomized controlled trials [12–15] and the qualitative study [2] mentioned in introduction section? More information about the necessity and importance of this study should be added.

3. Regarding the methodology, the realist evaluation seemed to have been done correctly. But in my opinion, the rationale for using this method may need to be strengthened, as it was not clear why this method is more suitable for analyzing the underlying processes of befriending interventions than other methods. It would be helpful to pay more attention to explaining why this method was used and what its strengths are.

4. The authors reported the inclusion criteria of participants, but they did not report in which families both service users and family members of service users were interviewed. It would be better to report this additional information.

5. Although “Rigour” section mentioned that other members of the research team reviewed the data collection and analysis process, much more information related to the reliability information of coding, such as Inter-rater Reliability of coding of one sample transcription (if available), is needed.

6. This study identified four mechanisms (i.e., reciprocity, empathy, autonomy, and privacy) in the analysis. I was wondering about the association of these mechanisms and contextual layers introduced in the introduction section. It seemed four mechanisms may belong to different layers. And what is the potential relationship between these four mechanisms? Some further discussion may be necessary.

Reviewer #2: The manuscript clearly identifies the link between psychological factors and interventions, which reduce loneliness and befriending. However, I would like to see a table that describes the findings with different categories; type of interventions (mechanisms) and effectiveness of interventions in various categories. This will improve the readability and the message or findings of the paper much easier.

6. PLOS authors have the option to publish the peer review history of their article (what does this mean?). If published, this will include your full peer review and any attached files.

Reviewer #1: No

Reviewer #2: No

---

## [Author Response · Author response to Decision Letter 0]

16 Aug 2021

Reviewer 1

1. The introduction section felt somewhat short though the processes of realist evaluation needed to be elaborated in the Method and Results (i.e., more than 16 pages for Method and Result but only 3 pages for the Introduction). Although this suggestion is purely stylistic, I would expect to get enough background information in the Introduction section. Specifically, I was kind of confused about the term “older people”, and I found myself wondering about the definition of this population in the current study. Moreover, there was also limited information about “loneliness” and “isolation” which are keywords of this manuscript. Therefore, I would suggest that author(s) highlight the importance of alleviating “loneliness” and “isolation” in older people.

Response: Thank you for your comment. We have included more information on the importance of alleviating loneliness and isolation as seen in the Introduction section on page 3, line 59, ‘‘There is increasing awareness about the potential harm that loneliness and social isolation can cause [1]. Loneliness can be defined as an undesirable subjective experience, related to ‘unfulfilled intimate and social needs’ [2] whereas social isolation can be understood as an objective concept capturing the absence of relationships, contacts or ties with other people [3]. Although loneliness can occur at all ages, it is a particularly common problem among populations of older people [4] as the opportunities for social contact are limited by various factors such as death of peers, physical limitations such as sensory deficits that limit communication, and/or mobility limitations that restrict their ability to visit family and friends [5]. Loneliness and social isolation have been associated with a range of health conditions and studies have demonstrated how they can lead to adverse physical and mental health consequences, especially among the older population [6,7].’’

Additionally, regarding defining the age range for the older population, we reported in the introduction on page 4, line 77, that ‘‘While befriending services are offered to a diverse range of populations of all ages and needs [11], this study focuses on support for older people. There are various definitions of the age range of ‘older’ populations [12] however most services broadly define older individuals as aged ‘‘50+ or 60+’’ p.17 [13].’’ Moreover, in the eligibility criteria of the study (page 10, line 203), we highlighted that ‘‘To be eligible, it was not required that services specified the age-range of its service users, only that they identified themselves as offering the service to older people.’’

2. This study aimed to uncover contextual variation and underlying mechanisms of befriending services and address the gap between prior researches. However, it seems still unclear how your study extended past work in this area. It would be important to be more specific. For example, what are “inconsistent findings” on the effectiveness of befriending? What are the brief results of randomized controlled trials [12–15] and the qualitative study [2] mentioned in introduction section? More information about the necessity and importance of this study should be added.

Response: Thank you for your comment. This has been addressed as more information on the findings from the previous studies have been reported as seen on page 4, line 89: ‘‘Previous research on befriending has predominantly been outcome-focused with aims to assess its effectiveness for improving psychological health outcomes using randomised controlled trials (RCTs) [21–24]. Mead, et al. [19] conducted a systematic review to identify the clinical effectiveness of befriending as a social intervention, particularly focusing on individuals who were experiencing depressive symptoms or emotional distress. Similar to existing literature [25], findings highlighted that befriending had a ‘modest but significant effect’ on depressive symptoms in the short-term [19]. In another systematic review of RCTs and quasi-experimental trials of befriending interventions [17], evidence of an overall improvement benefit in patient-reported primary outcomes among people with physical and mental health conditions was reported, but the effect size was small and not statistically significant. Authors concluded that the current evidence base does not allow for firm conclusions on the effects of befriending on specific outcomes [17].’’ 

Regarding the qualitative research, it was reported on page 5, line 107, that ‘‘On the other hand, with the exception of Lester, et al. [9], previous qualitative research on befriending has focused predominantly on identifying the experiences of the befrienders and/or service users [28–33]. Studies have identified common challenges such as the commitment required and benefits including the development of genuine relationships, companionship [32], mobility and engagement in various activities [33]. Qualitative research has not sought to identify or provide information on the mechanisms that produce the outcomes observed in befriending interventions such as the alleviation of loneliness and/or social isolation.’’ 

3. Regarding the methodology, the realist evaluation seemed to have been done correctly. But in my opinion, the rationale for using this method may need to be strengthened, as it was not clear why this method is more suitable for analyzing the underlying processes of befriending interventions than other methods. It would be helpful to pay more attention to explaining why this method was used and what its strengths are.

Response: Thank you for your comment. The limitations of experimental methods were addressed on page 5, line 101: ‘‘A weakness of the experimental format, particularly relevant to the evaluation of befriending services, is the ‘Martinson problem’ [26], whereby inconsistent findings have been produced on the effectiveness of befriending because of neglect of contextual variation brought about by aggregate-level (between group) analyses, and failure to embrace the complexity of befriending services or address mechanisms of change [17]. Another limitation of the experimental format is that it does not place emphasis on how or why programs work or fail [27].’’ Moreover, it was stated for qualitative research (seen on page 5, line 112) that ‘‘Qualitative research has not sought to identify or provide information on the mechanisms that produce the outcomes observed in befriending interventions.’’ The suitability of a realist evaluation was reported on page 6, line 123, where it was stated that ‘‘Befriending interventions are highly contextual as they are introduced within complex social systems, comprised of an interplay of individual, interpersonal and institutional characteristics, and the wider infrastructural system (see table 1)… All befriending interventions are conditioned by the action of layer upon layer of contextual influences, hence are in constant transformation. Therefore, evaluation of such interventions need to consider the settings (context) within which it is implemented. Contextual contingencies are likely to influence implementation success and failure, how the intervention achieves impact, why their impacts vary and also the extent to which befriending interventions can be successfully transferred from one context to another [37]. In this research, a realist evaluation, underpinned by the philosophy of scientific realism, was utilised to identify the intricate relationships and underlying processes of these interventions, using five case studies of befriending interventions, the study addresses some of the limitations of previous literature summarised above by focusing on contextual variation and the identification and action of mechanisms.’’

4. The authors reported the inclusion criteria of participants, but they did not report in which families both service users and family members of service users were interviewed. It would be better to report this additional information.

Response: Thank you for your comment. This information is in the results however to make things clearer, we have stated on page 14, line 286, ‘‘The total sample (n=37 female, n=9 male) across all five cases comprised of: service managers/coordinators (2 service managers and 4 service coordinators), befrienders (n=17), service users (n=14), and family members related to the service users interviewed (n=9).’’ 

5. Although “Rigour” section mentioned that other members of the research team reviewed the data collection and analysis process, much more information related to the reliability information of coding, such as Inter-rater Reliability of coding of one sample transcription (if available), is needed.

Response: Thank you for your comment. In regard to the rigour of the study, more clarity has been provided on the coding process of the transcripts as seen on page 13, line 270 where it is written that A number of strategies were used to address quality and rigour of the research. For example, two methods of data collection were used to ensure triangulation, and a clear audit trail was established through audio recordings, emails and interview notes. Service documents obtained were reviewed to identify relevant information to inform and support findings from the interviews. Moreover, recruitment of participants and interviews were carried out until consistent patterns were observed in the analysis which demonstrated that data saturation had been reached. Furthermore, the researcher coded initially the first three transcripts and the codes derived were reviewed by the primary supervisor (NMC) and discussed to develop joint understanding of what constitutes a context, mechanism and outcome in befriending interventions. Subsequent transcripts were coded independently by the researcher and both supervisors of the research project (NMC and MD) reviewed and oversaw the data collection and analysis process.’’ Moreover, the process to data analysis in realist evaluation is iterative and was a constant and evolving process between the researcher and the second author (NMC). Additionally, there is no guidance on inter-rater reliability in the RAMESES (Realist And Meta-narrative Evidence Syntheses: Evolving Standards) guidelines for realist evaluation (https://bmcmedicine.biomedcentral.com/articles/10.1186/s12916-016-0643-1). 

6. This study identified four mechanisms (i.e., reciprocity, empathy, autonomy, and privacy) in the analysis. I was wondering about the association of these mechanisms and contextual layers introduced in the introduction section. It seemed four mechanisms may belong to different layers. And what is the potential relationship between these four mechanisms? Some further discussion may be necessary.

Response: Thank you for your comment, it was very useful. This information has been added to the manuscript as seen on page 28, line 634: ‘‘The revised programme theories (appendix S7) explain how these four mechanisms (reciprocity, empathy, autonomy and privacy) were triggered in particular contexts to produce the outcomes observed. Within these theories, different contextual layers are represented. For example, regarding the individual capacity of key actors such as service users and befrienders, findings from this study highlighted how the health, mobility, motivation and capabilities of these individuals can impact the success of the befriending relationship. In regard to interpersonal relationships, befrienders who felt supported and received appropriate training and supervision from service managers reported benefits to their relationship with service users. With regards to the institutional setting, characteristics such as monetisation of befrienders and the short-term/long-term nature of the befriending relationship impacted outcomes. Future research should examine the influence of the wider-infrastructural system. Findings from this research highlighted that geographical proximity between the befriender and service user could influence the establishment of a relationship between both parties.’’

Reviewer 2

The manuscript clearly identifies the link between psychological factors and interventions, which reduce loneliness and befriending. However, I would like to see a table that describes the findings with different categories; type of interventions (mechanisms) and effectiveness of interventions in various categories. This will improve the readability and the message or findings of the paper much easier.

Response: Thank you for your comment. We are not clear on what the reviewer is asking for specifically. This paper does not aim to address the effectiveness of befriending but rather to uncover the mechanisms underpinning these interventions and identify how the work, for whom, in what circumstances and why. There are further tables in the appendix that elicit the programme theories that describe how these interventions work as well the characteristics of the befriending services involved in this study. Moreover, reviews of effectiveness of interventions already exist (e.g. Siette J, Cassidy M, Priebe S. Effectiveness of befriending interventions: a systematic review and meta-analysis. BMJ Open 2017;7:e014304. doi:10.1136/bmjopen-2016-014304) as well as reviews of the categorisation of interventions (e.g. Fakoya, O.A., McCorry, N.K. & Donnelly, M. Loneliness and social isolation interventions for older adults: a scoping review of reviews. BMC Public Health 20, 129 (2020). https://doi.org/10.1186/s12889-020-8251-6).

---

## [Editor Report · Decision Letter 1]

18 Aug 2021

How do befriending interventions alleviate loneliness and social isolation among older people? A realist evaluation study

PONE-D-21-14257R1

Dear Author,

We’re pleased to inform you that your manuscript has been judged scientifically suitable for publication and will be formally accepted for publication once it meets all outstanding technical requirements.

Kind regards,

Marcel Pikhart

Academic Editor

PLOS ONE
---

## [Editor Report · Acceptance letter]

27 Aug 2021

PONE-D-21-14257R1 

How do befriending interventions alleviate loneliness and social isolation among older people? A realist evaluation study  

Dear Dr. Fakoya:

I'm pleased to inform you that your manuscript has been deemed suitable for publication in PLOS ONE. Congratulations! Your manuscript is now with our production department. 

Kind regards, 

on behalf of

Dr. Marcel Pikhart 

Academic Editor

PLOS ONE